# Modularized Genes in an Adrenal Pathway Reveal a Novel Mechanism in Hypertension Pathogenesis

**DOI:** 10.3390/ijms26083782

**Published:** 2025-04-17

**Authors:** David W. Deng, Annie Ménard, Alan Y. Deng

**Affiliations:** Department of medicine, Université de Montréal, Montréal, QC H3T 1J4, Canada

**Keywords:** aldosterone synthesis, cardiac and renal function, CUEDC1, polygenic hypertension

## Abstract

Human epidemiological studies have statistically localized a multitude of quantitative trait loci (QTLs) for blood pressure (BP). However, their potential pathogenic mechanisms causing hypertension remain mysterious. To fill this void, we utilized congenic knock-in genetics to physiologically analyze the BP effects of individual and combinational QTLs. The effect magnitude from a single QTL in vivo ranged from 33.8 to 59.8%. ‘Double’ and multiple combinations of QTLs exhibited the same BP impact as a single QTL alone. Consequently, the products of these QTLs seemed to belong to the same pathway involved in physiological BP regulations. From this, we identified a novel pathway of hypertension pathogenesis in vivo controlled by the CUE domain containing 1 protein (Cuedc1). This pathway physiologically modulates blood pressure, aldosterone production, and renal and cardiac functions. CUEDC1 originated from common mammalian ancestors, partly explaining similar blood pressures between humans and rodents on this shared mechanistic basis. A translation of CUEDC1 into diagnostic and treatment applications to humans seems individualized and mechanistic because humans and rats may utilize the same BP-regulating mechanisms involving CUEDC1. The future sustainability of post-GWAS will depend on a balanced and robust ‘ecosystem’ provided by model studies that are founded on the physiologies and mechanisms of BP regulations in vivo.

## 1. Introduction

### 1.1. Background and Questions Raised

Large-scale human genome-wide association studies (GWAS) have statistically located nearly 1000 potential quantitative trait loci (QTLs) associated with blood pressure spreads in populations by single nucleotide polymorphisms (SNPs) [1]. A genome-wide survey combining population-based data with candidate cells has selected potential regulatory SNPs using polygenic risk scoring and Mendelian randomization [2,3,4]. From these, certain candidate genes and even drug targets have been chosen for clinical trials against hypertension [2,3]. However, these SNPs are mostly found in non-coding genome regions with unknown functions. Their appearance here seems to be the result of primate evolution independent of physiological controls of blood pressure [5]. Thus, these SNPs are primarily genome markers for potential QTLs nearby, not QTLs themselves.

Post-GWAS challenges are to test the outcomes of statistics and the association of localized QTLs by calibrating them against a BP-regulating physiology in causation. As the ultimate proof, molecularly identifying causal genes is essential to physiologically changing blood pressure, rather than being solely associated with spreads in populations. This direct in vivo evidence further allows us to mechanistically establish an etiology of hypertension pathogenesis well in advance of its underlying physiology [6,7]. Even more daunting is to understand the mechanisms for 1000 potential human QTLs that physiologically control blood pressure not merely individually but together [8], beyond polygenic risk scoring by statistics [2,3].

### 1.2. Revealing Mechanisms of Blood Pressure Regulation in Physiology

The paradigm pivot from polygenic risk scoring in epidemiology to physiology seems necessary in a post-GWAS era. Physiologically, polygenic QTLs that determine blood pressure in mammals, including humans and rodents, appear to be governed by a framework of modularity in genetic terms [9] and multi-steps in a pathway in mechanistic terms [8,10,11]. Within a pathway, a QTL product acts as a step upstream or downstream of another QTL product [8,10,11], but does not necessarily play a role as a BP-impacting agent, unless it is at the end stage of the pathway. For example, muscarinic cholinergic receptor 3 (M3R) signaling is a pathway [7] probably upstream of adrenergic signaling [12,13].

The renin–angiotensin–aldosterone system [14] is the best-known pathway and constitutes sequential steps, with angiotensin II acting at the end of the pathway, mediated by receptors that physiologically control blood pressure. Without knowing physiology, genome-scale studies using SNPs have implicated a wide range of previously unknown pathways. Our challenges are to physiologically uncover these pathways and dissect their hierarchy and regulatory relationships in vivo. The M3R signaling pathway is one such recent example.

### 1.3. Objectives

Our current work intends to further prove this paradigm-shifting concept in mechanistically understanding blood pressure as a polygenic trait. Driven by the physiology of BP regulations, we aimed at, first, establishing a BP physiological effect of a QTL for multiple QTLs. Second, we analyzed the combinational effects of two or more QTLs in physiologically regulating BP. Third, we identified a novel adrenal pathway that can alter blood pressure physiologically along with related phenotypes. Finally, we provided the evolutionary and mechanistic evidence that a newly discovered adrenal pathway of BP regulation originated from the common ancestors of humans and rodents. This pathogenic mechanism has the potential to innovate a personalized diagnostic and treatment target against human hypertension in a polygenic context. Appendix A outlines our approaches in a flow chart.

## 2. Results

### 2.1. Congenic Knock-Ins Mediate a Transition of Associative Results to Molecular Mechanisms Determining BP Physiology

Our initial studies linked/associated a segment of DSS Chromosome 10 (Chr. 10) with BP total variance. Since it is cost prohibitive to target multiple genes simply based on statistics and association results, we had to choose the most promising yet limited genes that may physiologically alter BP. We initiated this process with congenic knock-in genetics (Figure 1). This approach identifies only the QTLs that can alter BP in physiology [8,10,11], and in the process, it downgrades the chromosome segments not containing QTLs (e.g., C10S.L19 in Figure 1).

From the list of all rat chromosome regions containing QTLs [9], we focused on a single segment of DSS rats. Our work succeeded in isolating non-overlapping sections containing four distinct individual QTLs that physiologically regulate BP. The congenic knock-ins resolving each of the four QTLs reached experimental limitations based on scoring for crossovers.

### 2.2. Physiologically, a QTL Has a Major Effect on Blood Pressure

Epidemiologically, a QTL is believed to have a tiny effect on BP when fractionating it from total variance, which represents BP spreads in subject populations [1]. However, physiologically, each of the four QTLs has a major effect on blood pressure (Figure 1, Table 1). For instance, we can calculate the physiological effect of C10QTL6 as follows: knocking in the segment containing hypertensive C10QTL6 alleles by normotensive alleles in C10S.L18 (Figure 1) lowered BP by 30 mm Hg in vivo. Thus, the physiological effect of the C10QTL6 reflects the percentage of BP that has been altered by the QTL. Namely, 30 mm Hg (BP lowered by C10QTL6)/80 mm Hg (total BP difference between two parental strains) = 37.5%.

By the same functional measure, we can calculate a physiological effect for each of the three remaining QTLs (Figure 1). Consequently, each of the four QTLs alone has a BP effect ranging from 34.8 to 59.8% in vivo (Table 1). Thus, in calculating the effect of a given QTL by fractionating it from the total variance in blood pressure spreads is not equivalent to its physiological impact.

### 2.3. Three QTLs Act by Modularity/Common Pathways in Physiologically Controlling Blood Pressure

The above results beg the question: what is the combined physiological effect between two of these QTLs, as well as when all three QTLs are put together? This issue is important to address in the context of polygenic traits because an often-used statistical parameter in measuring blood pressure effects in epidemiology is multiple QTLs which would additively accumulate effects [1,15]. It appears that the most direct way of testing this hypothesis is to combine one QTL with another and observe the BP changes in vivo. As far as we know, no such physiological evidence is available from human GWAS and follow-up studies.

To address this issue, we systematically built and measured blood pressure changes in QTL combinations between two adjacent QTLs and all three QTLs (Figure 1). Congenic knock-in by C10S.L20 combined *C10QTL6* and *C10QTL1* lowered BP by 34 mm Hg, versus by 30 mm Hg from normotensive *C10QTL6* alleles alone, and by 50 mmHg from normotensive *C10QTL1* alleles alone, respectively (Figure 1). Obviously, the combined effect of *C10QTL6* + *C10QTL1* did not exceed that for each of the two QTLs individually (Figure 1); namely, epistasis [9] exists between *C10QTL6* and *C10QTL1* (*p* < 0.001). The similar epistasis of one gene hiding the effect of another was also seen between *C10QTL1* and *C10QTL5* in the ‘double’ congenic knock-in by C10S.L32. Furthermore, the combined effect of three QTLs together in the ‘triple’ congenic knock-in of C10S.L17 did not surpass that for each of the three QTLs alone. Thus, the three QTLs perform their functions in physiologically regulating BP by epistatic modularity [9]. The molecular basis of epistatic modularity is a common pathway in which each QTL product acts at one step up or down from another QTL product [8,9,10,11]. M3R is an example of such a step in a pathway [7,12], which is separate from the pathway integrating the three QTLs in our current work.

### 2.4. Revelation of a Pathway via Molecularly Identifying a QTL

To pinpoint a step in a pathway, the molecular identification of each of the four QTLs is necessary. One gene in each of these four QTL-residing regions is expected to be the causal gene for the QTL in question physiologically. This is based on the principle of one QTL (e.g., C17QTL1) being equal to one gene (e.g., Chrm3) in the polygenic context [7].

QTL identification at the physiology level relies primarily, although not exclusively, on identifying a gene whose mutation can affect a pathway leading to BP control. A commonly advocated approach seems quantitatively intuitive in identifying differentially expressed genes to find a QTL candidate [15]. However, there is no evidence that changing the level of expression in genes can actually alter a pathway and, in turn, blood pressure in vivo [12,16]. In fact, available experimental data showed that a QTL/gene dose variation did not cause a change in BP physiologically [12,16]. Instead, a missense mutation has been shown to alter the M3R signaling and lead to identifying *Chrm3* as *C17QTL1* in vivo [7,12]. From that evidence, we focused on screening missense mutations as primary leads to identify candidate genes for the QTLs in question, without eliminating differential gene expressions among them. None of these genes are known to influence BP. Thus, the identification of these QTLs will unveil novel physiological mechanisms determining BP.

Three candidate genes of a multi-gene Schlafen family for *C10QTL4* were found to carry at least one missense mutation (Table 1). As it stands, all three of these Schlafen genes are equally good candidates for *C10QTL4*. Candidate genes for *C10QTL1* and *C10QTL5* were found in our previous work [5].

### 2.5. The Gene (Cuedc1) Encoding CUE Domain Containing 1 Protein Is the Strongest Candidate for C10QTL6 in Hypertension Pathogenesis

First, fine resolution by congenic knock-in has narrowed the *C10QTL6*-residing interval to contain two genes, *Cuedc1* and *mitochondrial ribosomal protein S23* (*MrpS23*) (Figure 1). This is because both congenic knock-ins by C10S.L18 and C10S.L33 lowered blood pressure in vivo (Figure 1). *C10QTL6* should be located in the overlapping segment ˂170 kb between these two strains, which lodges two genes, *Cuedc1* and *MrpS23*. This strategy of localizing QTLs physically and physiologically is well established from our previous work [7,9].

Between *Cuedc1* and *MrpS23*, only *Cuedc1* possesses a single missense and significant mutation (Table 1). This molecular evidence identified *Cuedc1* to be the single candidate gene likely responsible for the function of *C10QTL6* in affecting BP. The caveats at the end of the Discussion Section address this issue further.

Second, *Cuedc1* is expressed in several organs, including adrenal glands, brain, kidneys, and the heart. Thus, these organs appear to be reasonable candidate organs for *Cuedc1*/*C10QTL6* to influence BP physiologically.

Third, as an index of kidney function, we analyzed creatinine clearance (Ccr). The Lewis *Cuedc1* alleles in C10S.L18 congenic knock-in ameliorated Ccr, i.e., 1.5 ± 0.1 mg/mL/g (kidney weight) versus 1.15 ± 0.05 mg/mL/g of the DSS parental control (*p* < 0.025, *t*-test). C10S.L18 congenic knock-in also normalized the diastolic dysfunction of DSS (Table 2), indicating that *Cuedc1* may impact cardiac function.

Fourth, we performed in situ hybridization with the *Cuedc1* antisense probe. *Cuedc1* is prominently expressed in the adrenal cortex (Figure 2). A higher magnification showed that the zona glomerulus is the primary site where *Cuedc1* is expressed (Figure 3). Coincidentally, the rate-limiting enzyme in aldosterone synthesis, aldosterone synthase, demonstrates a similar pattern of expression [14]. Thus, Cuedc1 likely participates in aldosterone synthesis/secretion from the zona glomerulus of the adrenal cortex.

Finally, substituting *Cuedc1* alleles of hypertensive DSS by those of normotensive Lewis reduced (*p* ˂ 0.02, *t*-test) the aldosterone production in adrenal glands (i.e., 29.35 ± 0.17 in ng/mg protein of C10S.L18, *n* = 7 versus 39.46 ± 0.28 in ng/mg protein of DSS, *n* = 6). Aldosterone in plasma in picomole/liter was also lowered (*p* ˂ 0.01, *t*-test) in C10S.L18 (i.e., 344.64 ± 18.20, *n* = 7) versus DSS (629.88 ± 2.46, *n* = 6). These data indicate that Lewis *Cuedc1* alleles may decrease aldosterone synthesis in the zona glomerulus and into circulation, and consequently lower BP in vivo. This functional impact occurs while genes encoding other enzymes and molecules in the aldosterone synthesis and secretion remain unchanged, such as *Cyp11b1* and *Cyp11b2*.

Cuedc1 is solely responsible for changing BP via regulating aldosterone synthesis, because a congenic C10S.L18 knock-in from Chr 10 (Figure 1) excluded two genes expressed in the zona glomerulus such as *Rgs4* on Chr 13 and *Dab-2* on Chr 2. C10S.L18 also rules out two genes known to be enzymes in the aldosterone pathway, *Cyp11b2* and *Cyp11b1*, both on Chr 7.

### 2.6. Direct Translation of the Hypertension Pathogenic Pathway of Cuedc1 from Rodents to Humans

A translation of *C10QTL6* identified in our rat models to humans is mechanistically direct in the physiology of BP regulation because *Cuedc1* originated from the common ancestors of humans and rodents. Despite no sequences being available from the extinct common ancestors of humans and rodents [17], the Cuedc1 pathway already existed in them. This is because the Cuedc1 protein of the Tasmania Devil, a marsupial, shares at least 87.9% of homology conservation with humans and rats overall (Table 3). This fact indicates that the Cuedc1 pathway in regulating blood pressure was present even in common ancestors of marsupials, placental rats, and humans (www.timetree.org). Marsupials have similar blood pressures to most placental land mammals including rats and humans [18].

This is proof that the BP-regulating mechanism for the Cuedc1 pathway originated from the common ancestors of humans and rats. Consequently, humans and rats inherit and then continue to employ the same Cuedc1 pathway in controlling blood pressure. No evidence exists that humans have lost this CUEDC1 mechanism. This physiological mechanism, in part, explains similar blood pressures between humans and rodents [18], whereas a possible convergent evolution has no proof of differing mechanisms before humans and rats diverged than to obtain similar blood pressures during evolution.

In contrast to the common origin of the Cuedc1 pathway in BP control, human non-coding GWAS SNPs associated with BP have nothing to do with BP physiology and only emerged during primate evolution, long after humans and rats diverged.

## 3. Discussion

### 3.1. Major Findings from Our Current Work

(a) Physiologically, each QTL can alter blood pressure by at least 33.8% of the total difference in mmHg between two parental strains. (b) The physiological impact of combining two QTLs or three QTLs does not exceed that of a single QTL alone. This epistatic outcome suggests that the three QTLs function in the same pathway to physiologically control blood pressure. (c) A previously unknown adrenal pathway of Cuedc1 has been discovered in pathophysiologically regulating hypertension in a polygenic context. (d) A human Cuedc1 pathway of hypertension pathogenesis is directly implicated, that is, predicated, on its shared origins of mechanisms in mammalian common ancestors in BP control.

### 3.2. Physiological Impact from a QTL on Blood Pressure Is Major

Assuming that a GWAS SNP truly indicates that a BP QTL is nearby, its potential effect on BP from GWAS is not predicated on how much it can alter BP physiologically, as shown in the current work (Table 1). Instead, this GWAS SNP’s effect is a percentage of the total variance [1], i.e., the extent of blood pressure spreads veering from their average mean in outbred populations. By this calculation, the SNP in question seemed to implicate a miniscule effect, deviating from the average value of blood pressure spread in the study population [1]. This is the essence of the ‘omnigenic’ hypothesis [15].

The actual blood pressure effect of a QTL in vivo has been evaluated by measuring its physiological change in BP in isolation and alone while maintaining the remainder of the genome as homogeneous and the environment as constant. As shown in the current work, the magnitude of the physiological effect from each of the four QTLs is at least 33.8% (Table 1). Another way of assessing whether an effect from a single QTL is miniscule or major physiologically is to eliminate that QTL in vivo. If the ‘omnigenic’ hypothesis [15] were physiologically valid, removing one single QTL among a multitude should have an invisible consequence on an organism. This is not the case, since *Chrm3* alone is responsible for *C17QTL1*, and eradicating it in vivo physiologically reduced BP by more than 50% in the BP difference between *Chrm3^+/+^* and *Chrm3^−/−^* [7].

### 3.3. Combined Effects from Multiple QTLs Are Physiologically Epistatic and Suggest a Shared Pathway Merging Them in Regulating Blood Pressure Physiologically

In one inbred rodent strain, there are over thirty QTLs, each of which changes blood pressure physiologically in a similar magnitude [9]. Thus, these QTLs seem physiologically redundant in accounting for the total BP difference. This phenomenon poses a physiological conundrum: what can their combined effects be? Apparently, adding them up will not work, because that would exceed the total blood pressure difference between two contrasting parental strains. The critical issue to address is not how ‘miniscule’ the BP effect from each QTL would be, but instead, why there are such overabundant QTLs that seem physiologically excessive in modulating BP.

When combined, two QTLs and even three have the same impact physiologically on blood pressure as a single QTL alone (Figure 1). This epistatic phenomenon is well known [8,9] and can account for the oversupply of QTLs in modulating blood pressure in an organism. Although epistasis is considered a nuisance in epidemiological studies [15], great mechanistic insights can be gained into a regulatory hierarchy and relationships among diverse components in a pathway that physiologically regulates blood pressure [10,11].

With the advent of genome-scale studies from both animal models [9] and humans [1], brand new physiological pathways have been unearthed that may be composed of far-flung steps away from the end-stage physiology of BP control [11]. This insight has emerged primarily from systematically studying epistasis among diversified QTLs [9,11]. From these epistatic analyses, a previously unknown pathway of M3R signaling in physiologically regulating BP has been revealed [7] that may act upstream of the adrenergic signaling pathway [12]. In physiologically controlling blood pressure, at least seven more pathways may participate upstream or downstream of the M3R signaling pathway that together constitute a grand pathway, known as epistatic model 2 [7,9].

### 3.4. A Novel Pathway of Hypertension Pathogenesis as Part of Polygenic Architecture Determining Blood Pressure Physiology

Another grand pathway parallel to epistatic module 2 is known as epistatic module 1, which affects the blood pressure physiology in additivity when combined with a member from epistatic module 2 [9]. Our current work represents a breakthrough in molecularly identifying a hypertension pathogenic pathway in epistatic module 1. In this context, one aspect of polygenicity in BP control arrives from the distinct Cuedc1 mechanism. At least 16 pathways up- or downstream of Cuedc1 are expected in the grand pathway contained in the module [9].

### 3.5. Discovering a New Role Played by Cuedc1 in Physiologically Controlling Blood Pressure via Regulating Aldosterone Synthesis and Kidney and Cardiac Functions

Our current studies produced the first evidence that the CUEDC1 pathway can physiologically control aldosterone production, renal and cardiac functions, and ultimately, blood pressure. Since aldosterone in both adrenal glands and plasma is affected, CUEDC1 likely participates in aldosterone synthesis rather than in its secretion into the circulation. A BP reduction caused by a drop in aldosterone by the Lewis Cuedc1 alleles is physiologically rational, thus mechanistically linking CUEDC1 to the BP homeostasis.

The mechanism by which CUEDC1 may influence the aldosterone synthetic pathway includes, but is not limited to, regulating the rate-limiting enzyme Cyp11b2 and/or catalyzing an biochemical step in the zona glomerulus [19]. Other possibilities cannot be ruled out that CUEDC1 might control substrate availability in mitochondria for Cyp11b2 to catalyze or it might affect the secretion of aldosterone from the zona glomerulus into circulation. Other functional roles of CUEDC1 at any level are less known, except for possessing a putative ERα-responsive enhancer located in the first intron of CUEDC1. ERα is a ligand-dependent transcription factor in cell proliferation [20].

Aldosterone is the principal steroid hormone that acts through mineralocorticoid receptors on the epithelial cells in the distal tubules and collecting ducts of the kidney to cause the retention of sodium and water, the secretion of potassium, and consequently, to increase BP [19]. Aldosterone also takes part in cardiac remodeling [21] and other functions [22].

Because Cuedc1 is required for body axis formation during embryogenesis [23], knocking it out cannot be viable in mammals, contrary to the case for *Chrm3* [7]. Since causal mutation occurs in DSS rats (Table 1), which are hypertensive, it is likely that CUEDC1 is pro-hypertensive. In contrast, *Cyp11b1* showcases an amelioration of hypertension independent of its pathogenesis, as the responsible mutations appear in normotensive Dahl salt-resistant rats (DSR) [24]. The normotension of DSR and Lewis rats is achieved via different pathways [25].

### 3.6. Direct Translation of CUEDC1 into a Human Mechanism in Polygenic Hypertension

Because humans and rodents have similar blood pressures [18], the mechanisms of blood pressure control from their common ancestors are unchanged in modern humans and rodents, despite their evolutionary divergence and size differences. This common origin in BP-modulating mechanisms can directly translate the results from our animal models to humans. Namely, the human CUEDC1 pathway should function in the same way as its rodent counterpart and plays roles in human BP control physiologically. There is no evidence that humans have disinherited this pathway. In this context, any unique human-centric mechanisms that appeared during primate evolution later on seem to account for nearly 0% of the total BP-regulating mechanisms of humans [5].

### 3.7. Human Medical Applications Can Be Generated in Personalized Diagnosis and Treatment of Hypertension by Targeting CUEDC1

In clinical applications, treating hypertension is an individualized task, not a mass delivery of one drug across heterogeneous populations in collectivity like a vaccine. The CUEDC1 pathway has become an individualized diagnostic and therapeutic target against human hypertension, even in a polygenic context. Regardless of their epidemiological prevalence, a thorough screening of *CUEDC1* coding mutations seems warranted among hypertensives in worldwide human populations. It is expected that a subset of human populations may carry a missensed *CUEDC1* mutation.

So far, no evidence from human cohorts and their follow-up in vitro studies has shown that any of the GWAS non-coding SNPs can change blood pressure physiologically. The simplest and most direct way to prove that a GWAS sentinel can affect blood pressure is to remove it from humans and to show a blood pressure change, such as the proof provided by *Chrm3* [7]. Obviously, this is a limitation in human studies.

Cellular models using CRISPR may be useful in addressing the causality issue for certain SNPs in vitro [4] but are limited in establishing BP-controlling mechanisms in physiology for them. Because hypertension occurred in vivo in spite of impaired vaso-dilation by muscarinic cholinergic receptor 3 (*Chrm3*) encoding M3R [6,7], inferring the BP outcome solely from damaged cellular vaso-dilation to a decrease in blood pressure would be misguided [6,7]. Systemic BP physiology is not the result of secluded and incidental cellular events but concerted and compensatory interconnections among multiple organs. There are no in vitro replacements for in vivo physiology proofs.

### 3.8. Multiple Steps of a Pathway Acting Together as a Module in Hypertension Pathogenesis

No human epidemiology markers were found near CUEDC1, partially due to an elevated prevalence threshold in the study populations for the GWAS, which set minor allele frequencies (MAF) ≥ 1% [1]. Markers lower than this MAF were filtered out and did not appear in the results. Nevertheless, the MAF setting is a pure epidemiological parameter for a given marker in a population that is unrelated to blood pressure regulations in physiology/causality.

Unlike well-known BP impacting materials such as angiotensin II, CUEDC1 by itself is not an end-stage product, immediately regulating the blood pressure physiology. Apparently, multiple steps upstream and downstream of CUEDC1 should exist in a pathway leading to an end stage in BP modulations [10,11]. The mechanisms of blood pressure control in each individual/inbred strain involve poly-pathways. CUEDC1 from the current work and M3R signaling from our previous work [7] are just two separate pathways found in one inbred polygenic model of hypertension. Numerous inbred rats of polygenic hypertension exist, and each consists of multiple pathways. Our knowledge of the mechanisms of blood pressure control in one single polygenic inbred model can be multiplied to a collection of inbreds until they cover the entire spectrum of an outbred human population.

### 3.9. Caveats

First, definitive proof is still lacking that *CUEDC1* is solely responsible for the physiological function of *C10QTL6*, since C10S.L18 contains more than just *Cuedc1*. Due to its involvement in embryogenesis, deleting *Cuedc1* from DSS rats by gene targeting is not viable, emulating the work on *Chrm3* [7]. In the future, we have to prove in vivo, by a *Cuedc1* knock-in specifically at 510T of the DSS rat, that the Serine by Arginine substitution (Table 1 and Table 3) alone can lower BP and aldosterone synthesis and improve renal and cardiac functions.

Second, in order to understand its molecular mechanisms, we need to uncover the placement of Cuedc1 in the aldosterone pathway and its mode of action. Because Cyp11b2 is the most critical enzyme in aldosterone synthesis taking place in adrenal glands, it is essential to elucidate if Cuedc1 can affect Cyp11b2 expressions, production, and activity in vitro and in vivo.

Third, within the same pathway toward BP control [10,11], it is not clear how *C10QTL1*/*Ppm1e* and *C10QTL5*/*Prr11* (Figure 1) can be integrated into a hierarchy in regulating, or being regulated by, the Cuedc1 function. Little molecular and cellular functions of Ppm1e and Prr11 have been characterized, even though both genes are also expressed in the adrenals [5].

Finally, appropriate case–control studies on CUEDC1 in humans may shed light on its potential relevance to human blood pressure control. A putative interaction network of CUEDC1 in the zona glomerulus can be examined to reveal how it may potentially interact with other components in the pathway of aldosterone synthesis. Particular attention will be paid to the potential involvement of Ppm1e and Prr1l in vivo. CUEDC1 protein functions need to be elucidated at all levels from molecular and cellular to the full body in the context of blood pressure control in physiology.

## 4. Materials and Methods

### 4.1. Animals

Protocols in animal manipulations were approved by our institutional animal committee (CIPA). All congenic strains used in the current study are depicted in Figure 1. At the bottom, blood pressures in mmHg are presented along with other parameters.

### 4.2. Mutation Screening and Verifications

First, coding regions and intron–exon junctions of all the genes in QTL-containing intervals (Figure 1) were obtained by our independent genomic and cDNA sequencings combined with a query of the Rat Genome Database (RGD) [26]. Following their detection, the chromosome segment carrying a given mutation was individually and independently validated (Table 1). Copy number variations were sought as genome deletions and duplications from the RGD [26].

### 4.3. BP Experimental Protocols and Analyses

Male rats were studied in congenic knock-ins. Breeding procedure, dietary treatments, telemetry implantation, postoperative care, and BP measurement durations were essentially the same as reported previously [8,10,11]. The repeated measures’ analysis of variance (ANOVA) followed by Dunnett’s test to compare parameters in MAPs between 2 groups and the power and sample size calculations were as given previously [8,10,11]. The *p* values given are the most conservative of all the data point comparisons during measurements.

### 4.4. Assessing Cardiac and Renal Functions and In Situ Hybridization

The same methods and protocols that were previously published [7,12] were used.

For cardiac measurements, transthoracic echocardiography was performed in rats sedated with isoflurane, using an S12 phased-array transducer and a standard echocardiographic Sonos 5500 system (Hewlett-Packard, Andover, MA, USA). The left ventricular (LV) M-mode spectrum was obtained from the parasternal short-axis view at the papillary muscle level. LV diameters were measured at both cardiac end-diastole (LVDd) and systole (LVDs), and LV fractional shortening (FS) was calculated as (LVDd–LVDs)/LVDd X 100%. The Teicholz method was employed to calculate LV volumes and the LV ejection fraction (EF). The thickness of the LV anterior (AW) and posterior wall (PW) at the end-diastole was also measured in this view, and the LV mass was calculated as [(LVDd + LVAW + LVPW)^3^ − LVDd^3^) × 1.04] × 0.8 + 0.14. LA M-mode data were also obtained at the aortic valve level in the parasternal long-axis view. LA dimensions at both cardiac end-diastole (LADd) and systole (LADs) were measured, and LA fractional shortening (FS) was calculated as (LADs − LADd)/LADs × 100%. Pulsed-wave (PW) Doppler was used to study transmitral inflow (TMF) in the apical 4-chamber view. The peak velocity of the early filling E wave, E wave deceleration time (EDT), deceleration rate (EDR), and time interval between mitral opening and closing (MD) were measured using TMF. PW Doppler was also used to study the trans-aortic outflow in the apical 5-chamber view. Aortic peak velocity, LV ejection time (ET: time interval from the beginning to the end of the aortic flow), and the time interval from aortic closing to opening (DD) were measured on the aortic flow PW Doppler spectrum. The LV global myocardial performance index (MPI) was calculated as (DD-MD)/LVET. Continuous wave (CW) Doppler was used to measure LV isovolumetric relaxation time (IVRT) at the conjunction of LV inflow and outflow in the apical 5-chamber view and corrected by the square root of the R-R interval [IVRT/(RR)^1/2^] on a simultaneously recorded ECG. The average of three consecutive cardiac cycles was used for each measurement.

For assessing renal functions, the male rats of each genotype were fed a 2% high-salt diet starting from 5 weeks of age for 7 weeks. Individual rats were put inside metabolic cages. Their urines were collected over a period of 24 h. The creatinine clearance, an indication of the renal excretory function, was analyzed by the hospital biochemical services at the CHUM for each genotype of rats. The calculation of creatinine clearance is as follows: Ccr = UCr × UV/1440 × Scr. Ccr, creatinine clearance; UV, volume urine; Scr, serum creatinine; 1440 = 24 h × 60 min.

For in situ hybridization (ISH), tissues were frozen-cut into 10 μm sections, mounted on glass microscope slides, fixed in formaldehyde, and hybridized with ^35^S-labeled cRNA antisense and sense probes, generating positive and negative (control) signals, respectively. Both cold and radioactivity-labeled cRNA probes were synthesized from DNA templates. After ISH, Cuedc1 gene expression patterns were analyzed by both X-ray film autoradiography (4-day exposure time) and emulsion autoradiography (14-day exposure time). Autoradiographic images come with their cresyl violet counterparts scanned from tissue sections. The microanalysis data are presented at both low and high magnifications.

### 4.5. Aldosterone Measurements

Aldosterone from adrenal glands was extracted by the methanol method [22]. Briefly, at 12 weeks of age, male rats were sacrificed by decapitation, and two adrenal glands from one rat were removed, cleaned, and homogenized in 100% methanol. After centrifugation to remove the debris, the supernatant was dried and dissolved in a PBS solution. Aldosterone in plasma and adrenal glands was measured by radioimmunoassay (RIA) by our standard hospital services.

## 5. Conclusions

In the overall landscape of complex traits, our results represent a significant advancement in identifying one QTL that changes blood pressure by physiology/causality. Insights are mechanistic and have opened the door for understanding a novel pathway by CUEDC1 in blood pressure regulation in a polygenic context. The human CUEDC1 pathway in hypertension pathogenesis has been implicated to originate from mammalian common ancestors and resulted in humans and rodents using the same pathway in blood pressure control. Direct, novel, and personalized diagnostic and therapeutic applications against human polygenic hypertension may be initiated by targeting CUEDC1. Our current findings epitomize translational medicines that, founded on unraveled hypertension pathogeneses, can create potentially non-existent yet desirable and personalized human diagnostic and therapeutic innovation against hypertension in a quantitative, complex, and polygenic framework.

## Figures and Tables

**Figure 1 ijms-26-03782-f001:**
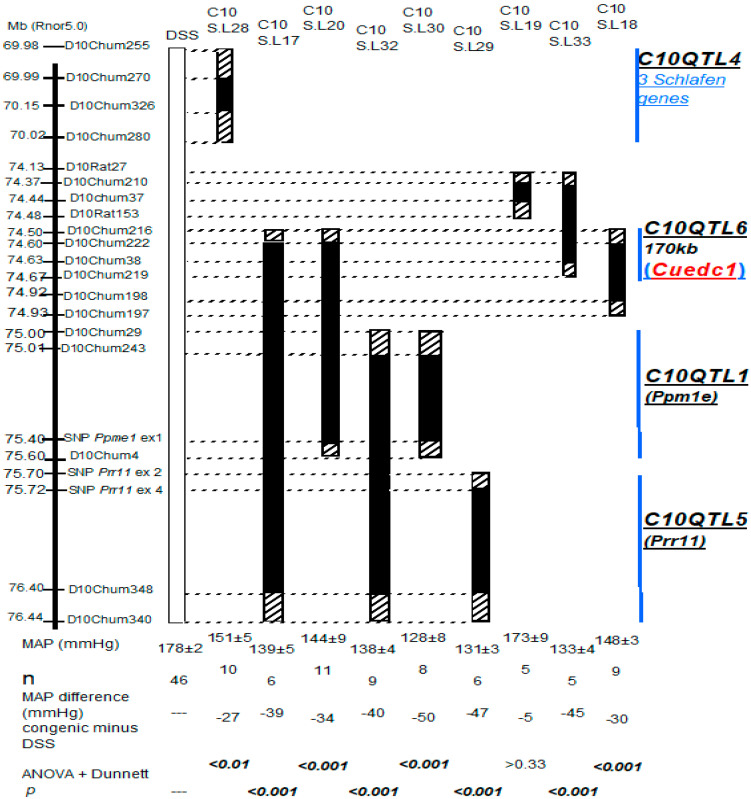
Congenic knock-ins defining 4 QTLs for physiologically changing blood pressure by causality. A solid bar under congenic strains with C10S.L prefixes represents the Dahl salt-sensitive rat (DSS) fragment that has been replaced (or knocked in) by its homolog of normotensive Lewis rats. C10S.L followed by a number designates a congenic knock-in strain in replacing a segment of S (DSS) by L (Lewis). Striped regions indicate the ambiguity of the crossover breakpoints between 2 markers. In designating a QTL, C10 refers to Chromosome 10, and a number after the QTL specifies a QTL to be defined. Mean arterial pressures (MAPs) for all the strains are averages for the duration of the measurement. Full gene names are listed in the footnote for Table 1.

**Figure 2 ijms-26-03782-f002:**
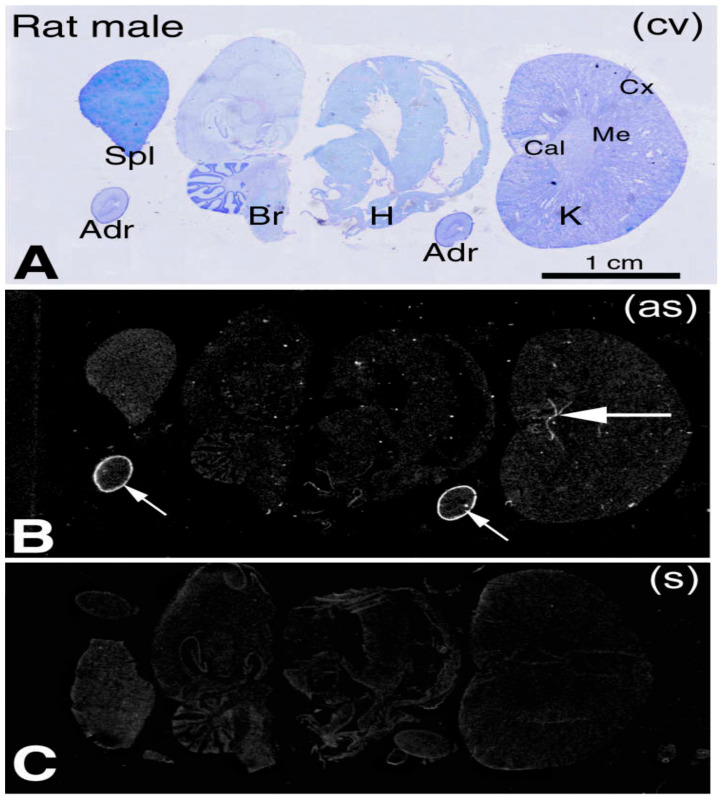
*Cuedc1* is specifically expressed in the adrenal cortex. Antisense (as) and sense (s) indicate that the in situ hybridization was performed with the respective *Cuedc1* probe. Staining and autoradiography of *Cuedc1* in selective rat organs are shown at a low magnification. Small and big arrows indicate the adrenal gland (Adr) and kidney calyx (Cal), respectively, where *Cuedc1* is expressed. (**A**) Cresyl violet (cv) staining; (**B**) *Cuedc1* as-probed; (**C**) s-probed. Br—brain; Cx—kidney cortex; H—heart; K—kidney; Me—kidney medulla; Spl—spleen. A similar pattern is seen with mouse organs.

**Figure 3 ijms-26-03782-f003:**
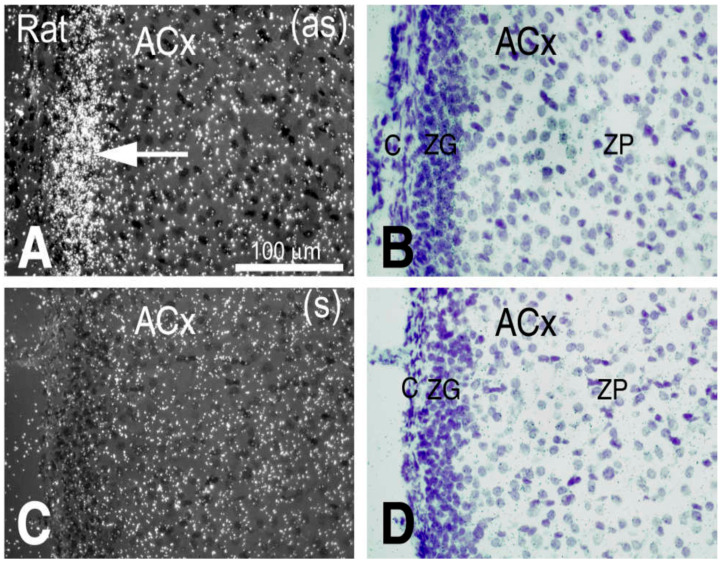
*Cuedc1* is expressed in the zona glomerulus (ZG). (**A**,**C**), autoradiography hybridized with *Cuedc1*-antisense (as) indicated by an arrow and sense (s) probes, respectively. (**B**,**D**), cresyl violet staining. ACx—adrenal cortex; c—adrenal capsule; ZP—zona pellucida or fasciculata. Data presented here are representative of several Dahl salt-sensitive rats.

**Table 1 ijms-26-03782-t001:** Candidate gene identification and assessment of a common mammalian origin.

C10QTLs	Blood Pressure Effect in Physiology	Gene	Mutation Detected Lew/DSS	Change in Amino Acid (AA) Lew/DSS	% Conservation in Amino Acid Comparing to Rat	
					Human	Tasmania Devil
** *C10QTL6* **	**37.5%**	** *Cuedc1* **	**G510T**	**A170S**	**88.9**%	**87.9**%
		*MrpS23*	none	None		
*C10QTL1*	62.5%	*Ppm1e*	1 missensed mutation			
*C10QTL5*	58.8%	*Prr11*	2 missensed mutations			
** *C10QTL4* **	33.8%	Schlafen 2 Schlafen 1 Schlafen 3 Schlafen 4 Schlafen 14	None Multiple Multiple G330C T1857C T3421C	None 6 amino acids 24 amino acids None None Tryp1141Arg		

**Footnote to Table:** The blood pressure effect for a given QTL was calculated by the percentage of blood pressure physiologically altered by the QTL. For *C10QTL6*, the blood pressure altered by the QTL is 30 mmHg (Figure 1). The total blood pressure difference between two parental strains, DSS and Lewis, is 80 mmHg. Thus, the actual effect is 30/80 = 37.5%. The same calculation applies to the other QTLs. The position of a mutation corresponds to the designation from the ATG start codon of that gene. The amino acid position begins with the first methionine. ***Cuedc1*,**
*CUE domain containing 1 protein*; ***MrpS23*,**
*mitochondrial ribosomal protein S23*. *C10QTL1* and *C10QTL5* mutations were identified in our previous work [5].

**Table 2 ijms-26-03782-t002:** Normalization of diastolic dysfunction in C10S.L18, which defines *Cuedc1*/*C10QTL6*.

Cardiac Phenotypes Measured by Echocardiography		DSS (*n* = 17)	Lewis (*n* = 13)	C10S.L18 (*n* = 10)
Heart rate (beats per min)		419.1 ± 28.0	**388.6 ± 34.4 ***	395.2.9 ± 37.8
Aortic peak velocity (cm/s)		107.2 ± 14.2	**94.6 ± 11.6 ***	**89.1 ± 9.9 ***
Left ventricular (LV) wall thickness and mass	LVAW (mm)	0.19 ± 0.02	**0.15 ± 0.01 ***	0.18 ± 0.02
	LVPW (mm)	0.19 ± 0.02	**0.15 ± 0.01 ***	0.17 ± 0.02
	Mass (g)	1.38 ± 0.09	**1.18 ± 0.13 ***	**1.23 ± 0.08 ***
	LV mass (g)	0.91 ± 0.09	**0.72 ± 0.13 ***	**0.88 ± 0.09 ***
LV hypertrophy and hyperdynamic state		Yes	**No**	**Normalized**
Left atrial (LA) dimension	Systolic (mm)	5.16 ± 0.63	**4.39 ± 0.48 ***	**4.6 ± 0.23 ***
	Diastolic (mm)	3.54 ± 0.62	**2.89 ± 0.38 ***	**2.92 ± 0.26 ***
	Fractional shortening (%)	31.4 ± 7.24	34.1 ± 6.08	36.7 ± 5.49
LA structural remodeling		Yes	**No**	**Normalized**
Pulse Doppler mitral filling pattern	E velocity (cm/s)	113.0 ± 14.9	**92.8 ± 12.7 ***	**102.3 ± 7.73 ***
	DT (ms)	45.9 ± 6.54	**59.3 ± 11.5 ***	53.2 ± 6.02
	DR (cm/s^2^)	2446 ± 460.6	**1548 ± 244.8 ***	**1914 ± 246.1 ***
Left ventricular isovolumetric relaxation time	IVRT (ms)	20.2 ± 3.12	**16.3 ± 2.66 ***	17.7 ± 4.58
	RR (ms)	144.3 ± 10.6	153.8 ± 12.9	153.0 ± 15.8
	IVRT/(RR)^1/2^	1.68 ± 0.25	**1.33 ± 0.23 ***	1.42 ± 0.30
Left ventricular global myocardial performance index	MD (ms)	53.8 ± 7.84	**62.0 ± 8.07 ***	61.2 ± 5.60
	DD (ms)	77.9 ± 9.63	79.3 ± 8.72	83.4 ± 10.6
	ET (ms)	70.5 ± 6.32	**79.4 ± 4.88 ***	**80.4 ± 6.15 ***
	MPI	0.35 ± 0.12	**0.22 ± 0.05 ***	**0.27 ± 0.03 ***
LV diastolic dysfunction		Yes	**No**	**Normalized**

Footnote: * and bold letters indicate *p* < 0.05 when compared to DSS. MD, mitral opening to closing; DD, aortic closing to opening; ET, ejection time; MPI, myocardial performance index; IVRT, isovolumetric relaxation time; RR, R-R interval; DT, E wave deceleration time; DR, deceleration rate; E velocity, E wave velocity; LVPW, left ventricular posterior wall thickness. Echocardiographic measurements are those described previously. The LV hyperdynamic state is based on aortic velocity data. LV diastolic dysfunction is based on the mitral filling pattern. LA structural remodeling is based on left atrial dimensions.

**Table 3 ijms-26-03782-t003:** Amino acid alignment for CUE domain containing 1 (*Cuedc1*) among humans, rats, and Tasmanian Devils.

DSS_Rat	MTSLFRRSSSGSGGGGATGARGAGTGTGDGSAAPQELNNSRPARQVRRLEFNQAMDDFKT	60
Human	MTSLFRRSSSGSGGGGTAGARGGG----GGTAAPQELNNSRPARQVRRLEFNQAMDDFKT	56
Tasmanian	MTSLFRRSSSNGGS------------RGGGNASAQELNNSRPARQVRRLEFNQAMEDFKT	48
	**********..*. .*.*: *********************:****	
DSS_Rat	MFPNMDYDIIECVLRANSGAVDATIDQLLQMNLEAGGVS--AYEDSSDSEDSIPPEILER	118
Human	MFPNMDYDIIECVLRANSGAVDATIDQLLQMNLEGGGSSGGVYEDSSDSEDSIPPEILER	116
Tasmanian	MFPNMDYDIIECVLRANNGAVDATIDQLLQMNLDG-----SSYDDSSDSDDSIPPEILER	103
	*****************.***************:. *:*****:**********	
DSS_Rat	TLEPDSSEEEPPPVYSPPAYHMHVFDRPYLTAPPTPPPRIDVLGSGQPASQSRYRNWNPP	178
Human	TLEPDSSDEEPPPVYSPPAYHMHVFDRPYPLAPPTPPPRIDALGSGAPTSQRRYRNWNPP	176
Tasmanian	TLEPDSSDEEPPPVYSPPAYHMHMFDRPYPLAPPTPPPRIDVPSAGVPLTQRRYRNWNPP	163
	*******:***************:***** **********. .:* * :* ********	
DSS_Rat	LLGNLPDDFLRILPQQMDSIQGHPGGSKPM-SGEGVPPVAPGPMACDQDSRWKQYLEDER	237
Human	LLGNLPDDFLRILPQQLDSIQGNAGGPKPG-SGEGCPPAMAGPGPGDQESRWKQYLEDER	235
Tasmanian	LLGNLPDDFLRILPQQLDSLQNTQSGPPKLGLGEVSQP---MVGNLEEECRWKQYLEDER	220
	****************:**:*. .* ** * :::.**********	
DSS_Rat	IALFLQNEEFMKELQRNRDFLLALERDRLKYESQKSKSSNVAVGSDVGFPSSVPG-----	292
Human	IALFLQNEEFMKELQRNRDFLLALERDRLKYESQKSKSSSVAVGNDFGFSSPVPG-----	290
Tasmanian	IALFLQNEEFMKELQRNRDFLLALERDRLKYESQKSKSSNMAVSNDFGFPSTVTGDAALG	280
	***************************************.:**..*.** * * *	
DSS_Rat	INDTNPTVSEDALFRDKLKHMGKSTRRKLFELARAFSEKTKMRKSKKKHLPKLQSLGAAA	352
Human	TGDANPAVSEDALFRDKLKHMGKSTRRKLFELARAFSEKTKMRKSKRKHLLKHQSLGAAA	350
Tasmanian	ASEANPAVSEDALFRDKLKHMGKSTRRKLFELARAFSEKTKMRKTKRKQLLKHQSAGWGL	340
	.::**:*************************************:*:*:* * ** * .	
DSS_Rat	STANLLDDVEGHAYEED--------FRGRRQEEPKVEE---------TLREGQ-------	388
Human	STANLLDDVEGHACDED--------FRGRRQEAPKVEE---------GLREGQ-------	386
Tasmanian	QHRQLISWMTWKAMRVKKTSGQGSRRHSRRRKHPEKDSKRCWSSPEMKCPNGQPNSEGLA	400
	. :*:. : :* . :.**:: *: :. :**	
DSS_Rat	-------------	388
Human	-------------	386
Tasmanian	LAAGTCCSKTEGV	413

* indicates amino acid identity (88.9%) between humans and rats, (90.4%) between humans and Tasmania Devils, and (87.9%) between rats and Tasmania Devils. Highlighted amino acids in color are mutated by comparing hypertensive DSS and normotensive Lewis rats. Humans and rats are placental mammals, whereas Tasmanian devils are marsupials. The human sequence was curated from https://www.ncbi.nlm.nih.gov/snp/, accessed on 30 December 2023.

## Data Availability

Data can be made available upon request to the corresponding author after publication.

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
