# Peer review of "Modularized Genes in an Adrenal Pathway Reveal a Novel Mechanism in Hypertension Pathogenesis"

_ijms, 2025, doi:10.3390/ijms26083782_

Round 1
Reviewer 1 Report
Comments and Suggestions for Authors
This is a very nice report describing a new role of Cuedc1 gene in blood pressure regulation and it’s association with high blood pressure in rats. The mechanisms behind this effect is obviously the role of CUEDC1 in the regulation of Aldosteron synthesis/secretion. I liked a lot the approach and the way data are presented. I only noticed that the methods section is far too short lacking some important information.
For example table 2 shows exciting echocardiography data but the respective methods are lacking. Also, the authors should check that all figures and tables state the exact number of animals used..
Author Response
Reviewer # 1:
This is a very nice report describing a new role of Cuedc1 gene in blood pressure regulation and it’s association with high blood pressure in rats. The mechanisms behind this effect is obviously the role of CUEDC1 in the regulation of Aldosteron synthesis/secretion. I liked a lot the approach and the way data are presented. I only noticed that the methods section is far too short lacking some important information.
For example table 2 shows exciting echocardiography data but the respective methods are lacking. Also, the authors should check that all figures and tables state the exact number of animals used.
Response to Reviewer # 1: Thanks for your appreciations for our `approach and the way data are presented`.
To address your concerns over methods section being too short, we have added more information on echo measurements as follows.
For cardiac measurements, transthoracic echocardiography was performed in rats sedated with isoflurane, using an S12 phased-array transducer and a standard echocardiographic Sonos 5500 system (Hewlett-Packard, Andover, Mass, USA). Left ventricular (LV) M-mode spectrum was obtained from the parasternal short-axis view at the papillary muscle level. LV diameters were measured at both cardiac end-diastole (LVDd) and systole (LVDs), and LV fractional shortening (FS) was calculated as (LVDd – LVDs) / LVDd X 100%. The Teicholz method was employed to calculate LV volumes and LV ejection fraction (EF). The thickness of the LV anterior (AW) and posterior wall (PW) at end-diastole were also measured in this view and LV mass was calculated as [(LVDd + LVAW + LVPW)3- LVDd 3) X 1.04] X 0.8 + 0.14. LA M-mode data were also obtained at the aortic valve level in the parasternal long-axis view. LA dimensions at both cardiac end-diastole (LADd) and systole (LADs) were measured and LA fractional shortening (FS) was calculated as (LADs – LADd) / LADs X 100%. Pulsed wave (PW) Doppler was used to study transmitral inflow (TMF) in the apical 4-chamber view. Peak velocity of early filling E wave, E wave deceleration time (EDT), deceleration rate (EDR), and time interval between mitral opening and closing (MD) were measured using TMF. PW Doppler was also used to study trans-aortic outflow in the apical 5-chamber view. Aortic peak velocity, LV ejection time (ET: time interval from the beginning to ending of aortic flow), and the time interval from aortic closing to opening (DD) were measured on the aortic flow PW Doppler spectrum. LV global myocardial performance index (MPI) was calculated as (DD-MD) / LVET. Continuous wave (CW) Doppler was used to measure LV isovolumetric relaxation time (IVRT) at the conjunction of LV inflow and outflow in the apical 5-chamber view, and corrected by the square root of the R-R interval [IVRT/(RR)1/2] on a simultaneously-recorded ECG. The average of three consecutive cardiac cycles was used for each measurement.
For assessing renal functions, male rats of each genotype were fed 2% high salt diet starting from 5 weeks of age for 7 weeks. Individual rats were put inside metabolic cages. Their urines were collected over a period of 24 hrs. The creatinine clearance, an indication of the renal excretory function, was analyzed by the hospital biochemical services at the CHUM for each genotype of rats. The calculation of creatinine clearance is as follows: Ccr = UCr x UV/1440 X Scr. Ccr, creatinine clearance; UV, volume urine; Scr, serum creatinine; 1440 = 24hr x 60 min.
For in situ hybridization (ISH), tissues were frozen-cut into 10-μm sections, mounted on glass microscope slides, fixed in formaldehyde and hybridized with 35S-labeled cRNA antisense and sense probes generating positive and negative (control) signals, respectively. Both cold and radioactivity-labeled cRNA probes were synthesized from DNA templates. After ISH, Cuedc1 gene expression patterns were analyzed by both x ray film autoradiography (4-day exposure time) and emulsion autoradiography (14-day exposure time). Autoradiographic images come with their cresyl violet counterparts scanned from tissue sections. The microanalysis data are presented at both low and high magnifications.
We hope that the added information is sufficient.
Also, we have double-checked the number of animals used in all figures and tables. All the numbers are real and exact.
Reviewer 2 Report
Comments and Suggestions for Authors
Dear Editor
The study by Deng et al., tried to examine novel loci for human blood pressure. They used congenic knock-in genetics in physiologically analyzing blood pressure effects of individual and combinational QTLs in male rats. They report that a novel pathway of blood pressure pathogenesis in vivo regulated by CUE domain containing 1 protein (Cuedc1). This pathway physiologically regulates the blood pressure via the aldosterone production, kidney and heart functions. I have the following comments
-abstract
the authors stated that "A translation of CUEDC1 into diagnostic and treatment applications to humans is individualized and mechanistic, because humans and rats utilize the same BP-regulating mechanisms involving CUEDC1"
in my opinion this is too early as the role of this protein in blood pressure regulation need further studies, please see limitations I suggest to add at the end of this report.
-introduction
The authors should provide certain SNPs or loci that are linked to hypertension see this paper Lip , Padmanabhan, Genomics of Blood Pressure and Hypertension: Extending the Mosaic Theory Toward Stratification, Can J Cardiol, 2020 May;36(5):694–705. doi: 10.1016/j.cjca.2020.03.001.
- the authors should introduce the pathways regulating the blood pressure.
- No need to subdivide the introduction subtitles.
Materials and methods
-Results
In table 3, the authors stated the 170 amino acid residue of Cuedc1 in human is R (arginine). But in uniprot (https://www.uniprot.org/uniprotkb/Q9NWM3/entry#sequences) the 170 amino acid residue is Y (tyrosine).
-Discussion and conclusion
-the authors should show the interaction networks of Cuedc1 (e,g, with GENEmania), and discuss how Cuedc1 influences blood pressure.
-what is the functions of the CUEDC1?
- in my opinion a good way to proof the influence of CUEDC1 on blood pressure is case-control studies in different populations to examine the effect of its mutation. In addition to CUEDC1 protein functional studies as well as bioinformatics analysis. I thin this is not difficult since CUEDC1 is relatively a small protein (386 amino acid residues). I suggest to add these to the limitations of this studies.
- the authors should hypothesize a detailed mechanism on how CUEDC1 can regulate blood pressure
- The English needs revision
Author Response
Reviewer #2: The study by Deng et al., tried to examine novel loci for human blood pressure. They used congenic knock-in genetics in physiologically analyzing blood pressure effects of individual and combinational QTLs in male rats. They report that a novel pathway of blood pressure pathogenesis in vivo regulated by CUE domain containing 1 protein (Cuedc1). This pathway physiologically regulates the blood pressure via the aldosterone production, kidney and heart functions. I have the following comments
Reviewer # 2 comment on abstract: the authors stated that "A translation of CUEDC1 into diagnostic and treatment applications to humans is individualized and mechanistic, because humans and rats utilize the same BP-regulating mechanisms involving CUEDC1". in my opinion this is too early as the role of this protein in blood pressure regulation need further studies, please see limitations I suggest to add at the end of this report.
To further address your concern, we have added `may` in the abstract. Now it reads ` because humans and rats may utilize the same BP-regulating mechanisms involving CUEDC1`. also a limitation as you suggested has been added to the end of the manuscript in Caveat section.
Reviewer # 2 comment on introduction: The authors should provide certain SNPs or loci that are linked to hypertension see this paper Lip , Padmanabhan, Genomics of Blood Pressure and Hypertension: Extending the Mosaic Theory Toward Stratification, Can J Cardiol, 2020 May;36(5):694–705. doi: 10.1016/j.cjca.2020.03.001.
Response to Reviewer # 2 for introduction (1): In humans, more than 1000 loci and over 10000 SNPs have been associated with blood pressure. They have been listed in ref. 1. Which ones would you like us to provide? As discussed in Discussion, there are no SNPs close to CUEDC1. In the current research paper, we focus on CUEDC1.
Reviewer # 2 comment on introduction (2): the authors should introduce the pathways regulating the blood pressure.
Response to Reviewer # 2 for introduction (2): Good idea. A brief introduction has been added as follows:
`The renin-angiotensin-aldosterone system (14) is the best-known pathway and constitutes sequential steps with angiotensin II acting at the end of the pathway mediated by receptors physiologically controlling blood pressure. Without knowing physiology, genome-scale studies using SNPs have implicated a wide range of previously unknown pathways. Our challenges are to physiologically uncover these pathways and dissect out their hierarchy and regulatory relationships in vivo. The M3R signaling pathway is one of such recent examples`.
Reviewer # 2 comment on introduction (3): No need to subdivide the introduction subtitles.
Response to Reviewer # 2 for introduction (3): The journal recommended subdivision in introduction, results, discussion, and methods. In keeping with their recommendations, we have kept them, if you do not fiercely object. Its merely an editorial thing, not scientific one. I hope it`s ok with you.
Reviewer # 2 comment on Materials and methods: There are no comments made.
Response to Reviewer # 2 on Materials and methods : we have added more detailed descriptions in methods on cardiac, renal measurements and in situ hybridization.
Reviewer # 2 comment on Results: In table 3, the authors stated the 170 amino acid residue of Cuedc1 in human is R (arginine). But in uniprot (https://www.uniprot.org/uniprotkb/Q9NWM3/entry#sequences) the 170 amino acid residue is Y (tyrosine).
Response to Reviewer # 2 on results: We have found the sequence from https://www.ncbi.nlm.nih.gov/snp/. A sentence has been added to the legend of Table 3. The difference between uniprot and ncbi could be just a matter of polymorphisms among individuals in human populations. We have not sequenced any human CUEDCI genes to address this issue. We can not say it`s R or Y.
Reviewer # 2 comment on Discussion and conclusion (1): the authors should show the interaction networks of Cuedc1 (e,g, with GENEmania), and discuss how Cuedc1 influences blood pressure.
Response to Reviewer # 2 comment on Discussion and conclusion (1): Interaction networks of Cuedc1 that you are talking about is based on bioinformatics, not on physiology, and not on function. As far as we know, no adrenal networks of Cuedc1 in zona glomerulas are known that can physiologically regulate blood pressure. Our current work has provided the first physiological evidence. We are focusing on the physiological aspect of CUEDC1, rather than speculating on how it could control blood pressure without any physiological evidence.
How CUEDC1 influences blood pressure is discussed in 3.5 of the manuscript. Please see our response to your comment on Discussion and conclusion (4) later on.
Reviewer # 2 comment on Discussion and conclusion (2): what is the functions of the CUEDC1?
Response to Reviewer # 2 comment on Discussion and conclusion (2). Little is known about the function of CUEDC1. As presented in Discussion 3.5, its only known function is having a putative ERα-responsive enhancer located in the first intron of CUEDC1. ERα is a ligand-dependent transcription factor in cell proliferation. Another piece of evidence is that CUEDC1 is required for development.
Our current work has provided the first evidence that CUEDC1 is physiologically involved in regulating blood pressure via controlling aldosterone synthesis in zona glomerulas, renal and cardiac functions.
Reviewer # 2 comment on Discussion and conclusion (3): in my opinion a good way to proof the influence of CUEDC1 on blood pressure is case-control studies in different populations to examine the effect of its mutation. In addition to CUEDC1 protein functional studies as well as bioinformatics analysis. I thin this is not difficult since CUEDC1 is relatively a small protein (386 amino acid residues). I suggest to add these to the limitations of this studies.
In any case, association studies are not proofs of CUEDC1 functions in physiologically regulating blood pressure. As presented in the first caveat, an ultimate proof can only come from a knock in study in vivo. Our current congenic knock-in in vivo is the next best proof that Cuedc1 may regulate blood pressure physiologically. Our current work has provided new mechanistic insights into a novel pathway in hypertension pathogenesis in vivo mediated by Cuedc1. We hope that you would recognize the significance of our discoveries.
You may contend that the inference that the CUEDC1 pathway controls BP through aldosterone is not proven in the human just because is conserved between the rat and the marsupial. We hope to convince you that the inference of a common CUEDC1 pathway in modulating blood pressure is physiologically valid for following reasons.
Rats and humans along with most land mammals have similar blood pressures (Evolution. 2014,68:901). How can this be? They are different in many other physiological traits such as size, longevity, birth rate etc. What makes blood pressure so unchanged and primal across the mammalian class? The only reason for this to occur is that mechanisms in physiologically regulating blood pressure was present in common mammalian ancestors before humans, rats and other mammals diverged. Humans, rats and other mammals simple inherit the same mechanisms until this day.
An explanation in convergent evolution can not be the cause. Convergent evolution states that 2 organisms started a trait (e.g. blood pressure) differently in ancestors, but arrived at the same state of the trait by accident. This can not apply to blood presssure. This is because rats and humans might, by accident, achieve the same blood pressure. How about other placental mammals such as pigs, tigers, rabbits etc? They diverged at different times during mammalian evolution. How can they also obtain the same blood pressure as other placental humans and rats? Moreover, marsupials diverged from placental mammals a lot earlier during evolution. How can they have same blood pressure as placental mammals? The answer has to be that common mammalian ancestors have created blood pressure mechanisms that later humans and rats inherit and continue to use till this day.
In another word, when we examine blood pressure-controlling mechanisms, we look back at our mechanistic past before modern humans and rats even existed. There is an analogous situation in physics. Knowing the speed of light and the distance, whatever we see from the sun on earth happened nearly 7 mins ago in the past, not at the instance when we examine it. It is a scientific fact.
By the same token, Cuedc1 as a blood pressure-regulating mechanism was created not at the present in both humans and rodents, but rather at least 90 million years ago in the past in our common ancestors before modern humans and rodents even existed.
One may argue that common mammalian ancestors have gone extinct. Nobody has seen one alive. How do we know they actually existed millions of years ago, or they were abstract notions in evolutionary paleontology? In fact, there are lines of concrete evidence indicating that mammal ancestors did exist and diversified during evolution (Science 2013,339:662).
We hope these explanations make sense to you. We`d be happy to entertain any different points of view and explanations, if you have.
Reviewer # 2 comment on Discussion and conclusion (4): the authors should hypothesize a detailed mechanism on how CUEDC1 can regulate blood pressure.
Response to Reviewer # 2 comment on Discussion and conclusion (4). We have discussed this issue extensively in 3.5. If you have any new suggestions, we`d like to hear them.
Reviewer # 2 comment on Discussion and conclusion (5): The English needs revision. Response to Reviewer # 2 comment on Discussion and conclusion (5). We have run through editing and English phrasings.
Round 2
Reviewer 2 Report
Comments and Suggestions for Authors
-in my opinion this is too early to conclude that CUEDC1 regulates blood pressure and suggest that it can be used in diagnostic and treatment applications as as the role of this protein in blood pressure regulation need further studies.
-in my opinion it is better to examine the interaction network of CUEDC1 and try to see a relationship to blood pressure. A well designed cases controls study and CUEDC1 protein functions studies can be employed for this purpose.
-please include limitations to this study
Author Response
Reviewer comments: in my opinion this is too early to conclude that CUEDC1 regulates blood pressure and suggest that it can be used in diagnostic and treatment applications as as the role of this protein in blood pressure regulation need further studies. -in my opinion it is better to examine the interaction network of CUEDC1 and try to see a relationship to blood pressure. A well designed cases controls study and CUEDC1 protein functions studies can be employed for this purpose. please include limitations to this study
Response to Reviewer comments: We share your concerns about definitive proofs of CUEDC1 in regulating blood pressure, as elaborated in Caveat 1. We are in the process of performing the single nucleotide knock in.
We agree with you on limitations. We have added these limitations in the last paragraph of Caveat in the revised manuscript as follows:
`Finally, appropriate case-control studies on CUEDC1 in humans may shed light on its potential relevance to human blood pressure control. A putative interaction network of CUEDC1 in zona glomerulas can be examined to reveal how it may potentially interact with other components in the pathway of aldosterone synthesis. Particular attention will be paid to potential involvements of Ppm1e and Prr1l in vivo. CUEDC1 protein functions need to be elucidated at all levels from molecular, cellular to the full body in the context of blood pressure control in physiology`.
We hope these added limitations have addressed your concerns.
In spite of these limitations, our current studes provided the first physiological and mechanistic evidence that the CUEDC1 pathway can alter blood pressure, renal and cardiac functions, most likely via regulating aldosterone synthesis in zona glomerulas. These results give more impetus for further studies as outlined in caveats including the limitationns that you suggested. We hope you appreciate the significance of our results and discussions.
Like any significant discovery, there are always more questions than answers. We`d like to address further functional questions in our future search. Hopefully, our results, once published, will stimulate interests from other scientists, especially human geneticists.
Round 3
Reviewer 2 Report
Comments and Suggestions for Authors
Slight English revision is required.